# Reporting and Help-Seeking among Sexual Minority and Heterosexual Victims of Stalking

Jessica C. Fleming [1,*], Ashley K. Fansher [2], Ryan Randa [1] and Bradford W. Reyns [3]

1    College of Criminal Justice, Sam Houston State University, Huntsville, TX 77341, USA; ryan.randa@shsu.edu
2    Department of Criminal Justice, University of North Dakota, Grand Forks, ND 58202-8050, USA; ashley.fansher@und.edu
3    Department of Criminal Justice, Weber State University, Ogden, UT 84408, USA; breyns@weber.edu
*    Correspondence: jcc063@shsu.edu

**Abstract:** Given the disproportionately higher rates of stalking among sexual minority individuals, the present study aimed to explore factors that influence these victims' help-seeking behaviors. Employing data from the United States' 2019 National Crime Victimization Survey (NCVS) Supplemental Victimization Survey (SVS), this study explores various demographic and psychological factors impacting help-seeking among sexual minority and heterosexual victims. Results indicate that sexual minority individuals are significantly more likely to seek help than heterosexual victims of stalking. Further exploration through logistic regression, limited to the sexual minority group, shows significant associations between help-seeking and age, sex, and emotional distress from stalking, but not race. Indicating that younger respondents, female respondents, and those experiencing an emotional impact are more likely to seek help for stalking victimization among sexual minority victims. These findings emphasize the importance of sexual orientation in understanding help-seeking behaviors among stalking victims, suggesting a need for more tailored support services for the sexual minority community.

**Keywords:** stalking; victimization; help-seeking; sexual orientation; emotional impact; sexual minority

## 1. Introduction

Stalking involves a repeated course of conduct on the part of a perpetrator that harms or causes fear/distress in the victim (e.g., Fox et al. 2011; Nobles et al. 2014; and Tjaden and Thoennes 1998). Stalking was first criminalized in the United States approximately three decades ago and was since researched extensively (e.g., Brady et al. 2020; Fisher et al. 2002; and Gatewood Owens 2017), but gaps remain in the extant literature. The present study focuses on the aftermath of the crime, particularly whether victims of stalking decide to seek help following their experience. Victim decision-making emerged as an important area of focus within victimology, with researchers investigating victim decisions ranging from help-seeking, to self-protection, to bystander intervention (e.g., Coker et al. 2011; Fisher et al. 2003, 2007; Reyns and Englebrecht 2010, 2014; and Tark and Kleck 2004).

Generally speaking, victim help-seeking decisions involve formal or informal sources of help. Victims may formally invoke the criminal justice system by calling law enforcement to respond to the crime. This represents an important step in the criminal justice process and provides a gatekeeper role for crime victims (e.g., Gottfredson and Gottfredson 1990). Formal help-seeking may also afford victims opportunities to utilize victim services. Help-seeking can also be more informal and include reaching out to friends or family for assistance, support, or advice (e.g., Fisher et al. 2003; McCart et al. 2010; and Reyns and Englebrecht 2014). Overall, victim help-seeking decisions shape the course of events for the criminal justice system, as well as for crime victims, as they begin the recovery process.

From this perspective, the present study utilizes data from the most recent 2019 United States-based National Crime Victimization Survey (NCVS) Supplemental Victimization

Survey (SVS) to address three issues surrounding the help-seeking decisions of crime victims. First, we identify factors that influence the help-seeking decisions of victims of stalking. Prior research suggests that the determinants of formal and/or informal help-seeking are not uniform, and instead differ according to the type of crime that the victim experienced (e.g., Brady et al. 2023; Fisher et al. 2003; Reyns and Randa 2017; and Xie and Baumer 2019). Thus, a focus on stalking victims' decisions, in particular, may suggest best practices that encourage reporting as well as barriers to recovery.

Second, the present research investigates help-seeking decisions through the lens of sexual orientation, emphasizing the decision-making of lesbian, gay, bisexual, and queer/questioning (sexual minority) stalking victims (see Kinsey et al. 1998). Previous research, although limited, indicates that sexual minority persons experience stalking at higher rates than persons who are heterosexual (e.g., see Edwards et al. 2022, for a systemic literature review). Given the disparity in victimization rates across groups, presumably, there may also be a disparity in help-seeking behavior. Research suggests that help-seeking, formal or informal, is influenced by victim characteristics, such as sex, age, or race (e.g., Brady et al. 2023; Fisher et al. 2003; and Reyns and Randa 2017), but thus far, it is an open empirical question whether the victim's sexual orientation affects their help-seeking decisions.

Third, this study identifies the factors that impact help-seeking decisions according to sexual orientation. We seek to identify the determinants of help-seeking among sexual minority victims and compare them to heterosexual victims of stalking. Given the disparity in victimization rates observed in previous research, there also may be differences in the factors contributing to sexual minority and heterosexual stalking victims' decisions to seek help for their victimization. Overall, the current study explores disparities in stalking victimization experiences with respect to help-seeking between these groups and highlights the intersectionality of sexual orientation and these experiences. Thus, findings from this research may help policymakers and law enforcement officials better understand the differing needs of sexual minority and heterosexual stalking victims and devise appropriate strategies to help them.

## 2. Review of Literature

### 2.1. Stalking Victimization among Sexual Minority and Heterosexual Individuals

Stalking is a crime characterized by persistent, unwanted pursuit, often inciting fear and emotional distress in victims through experiences such as unwanted following or communication (Logan and Walker 2017; Morgan and Truman 2022; and Reyns et al. 2023). Stalking victimization affects a substantial number of individuals and remains a serious public health concern (Morgan and Truman 2022; Purcell et al. 2002; Reyns et al. 2016; Smith et al. 2022; and Spitzberg and Cupach 2007). Estimates show that in 2019, approximately 1.3% (3.4 million) of all individuals in the United States, 16 years of age or older were victims of stalking (Morgan and Truman 2022). Differences in the prevalence of stalking victimization exist between different demographics, such as sex, age groups, marital statuses, and household incomes, with younger individuals, divorced or separated individuals, and those in households annually earning less than USD 25,000 experiencing higher rates of stalking victimization (e.g., Elvey et al. 2018; Fisher et al. 2002; Morgan and Truman 2022; Mustaine and Tewksbury 1999; Purcell et al. 2002; Reyns et al. 2016; Smith et al. 2022; and Spitzberg and Cupach 2007). Moreover, national estimates from the United States' Bureau of Justice Statistics (BJS) and the Centers for Disease Control and Prevention (CDC) indicate that females are stalked at approximately twice the rate of males (Morgan and Truman 2022; Smith et al. 2022).

Extant research shows that both heterosexual and sexual minority populations experience stalking victimization (Boyle and McKinzie 2021; Cantor et al. 2019; Chen et al. 2020; Edwards et al. 2015a, 2015b; Langenderfer-Magruder et al. 2020; Mennicke et al. 2021; and Walters et al. 2013). However, assessing the prevalence of stalking victimization among sexual minority victims is challenging due to methodological differences across

studies. As a result, reported victimization rates among the sexual minority populations exhibit significant variation, ranging from 7% to 53% (Boyle and McKinzie 2021; Cantor et al. 2019; Chen et al. 2020; Edwards et al. 2015b, 2022; Langenderfer-Magruder et al. 2020; Nobles et al. 2018; Sheridan et al. 2019; Trujillo et al. 2020; Turell 2000; and Walters et al. 2013).

Despite an increasing understanding of the prevalence of stalking victimization, there is limited research on the differences in stalking victimization based on sexual orientation, with existing research finding conflicting results. Some studies indicate that sexual minority individuals face a higher risk of stalking victimization compared to their heterosexual counterparts (Edwards et al. 2015b, 2022; Langenderfer-Magruder et al. 2017; Mennicke et al. 2021; Sheridan et al. 2019; and Walters et al. 2013), other studies find no such differences (Boyle and McKinzie 2021; Langenderfer-Magruder et al. 2020; and Turell 2000). Data from several studies identified that bisexual individuals report a higher lifetime prevalence of stalking victimization than other sexual orientations (Chen et al. 2020; Langenderfer-Magruder et al. 2020; Nobles et al. 2018; and Walters et al. 2013). This is supported by Walters et al. (2013) using estimates from the National Intimate Partner and Sexual Violence Survey (NISVS); they reported that one in three bisexual women experience stalking during their lifetime, compared to one in six heterosexual women. These findings are consistent with the sexual minority domestic violence and sexual minority intimate partner violence research (Langenderfer-Magruder et al. 2017; Martin-Storey 2015; and Stults et al. 2021).

As noted above, other studies revealed no statistically significant variations in the rates of stalking experienced by sexual minority and heterosexual stalking victims (Boyle and McKinzie 2021; Langenderfer-Magruder et al. 2020; and Turell 2000). These contrasting findings, however, should be interpreted with caution, as they are drawn from studies that rely on community-based or college student samples and do not use nationally representative data (Boyle and McKinzie 2021; Langenderfer-Magruder et al. 2020; and Turell 2000). Moreover, these contrasting results may reflect outdated practices, given that some studies cited are based on data from over 20 years ago (Turell 2000). While Langenderfer-Magruder et al. (2020) found no significant differences in reporting behaviors of stalking victims based on sexual orientation, their study did find that sexual minority female victims reported stalking to the police at higher rates than male sexual minority victims and transgender victims. Additionally, the study found that lesbian victims reported stalking victimization to the police at higher rates than gay, bisexual, or queer victims.

### 2.2. Reporting and Help-Seeking Behaviors

After an individual is victimized, they are left with several decisions: they can report the incident to the police, confide in a friend or family member, seek help from other professional services, or even choose not to disclose their victimization to anyone. Studies consistently demonstrated that victims of crime tend to rely more on informal sources, such as family and friends for help, rather than on formal sources, such as the police, mental health services, or medical professionals (e.g., Barrett and St. Pierre 2011; Coker et al. 2000; Fisher et al. 2003; McCart et al. 2010; Reyns and Englebrecht 2014; Sylaska and Edwards 2014; Ullman 2007; and Xie and Baumer 2019). Furthermore, a crime victim's decision to seek help can vary based on a number of factors, including age, sex, race, and the relationship between the victim and offender (Kaukinen 2002, 2004; Sylaska and Edwards 2014; and Xie and Baumer 2019). Research also suggested that the type of victimization experienced may influence help-seeking decisions. For example, Reyns and Englebrecht (2014) found increased odds of victims engaging in formal and informal help-seeking for stalking victims who were also cyberstalked.

Research on the help-seeking behaviors of stalking victims is limited, but findings are comparable with other types of violence. According to a United States Bureau of Justice Statistics report, less than one-third (29%) of all American stalking victims reported their victimization to the police in 2019 (Morgan and Truman 2022). While stalking victims may turn to formal sources, such as law enforcement, when their attempts to seek informal



help fail to deter their stalker (Campbell and Moore 2011; Reyns and Englebrecht 2014; and Taylor-Dunn et al. 2021), the low rate of reporting suggests that many victims may not feel comfortable seeking help from formal sources. Studies show that stalking victims may choose not to report incidents to the police for a variety of reasons, including a lack of faith in the ability or willingness of police to help, perceiving the incident as minor or not important enough to report, or fear of retaliation from the offender (Baum et al. 2009; Fisher et al. 2002; and Tjaden and Thoennes 1998). However, research also found that stalking victims who experienced both traditional and cyberstalking were more likely to apply for a restraining, protection, or no-contact order compared to victims of traditional or cyberstalking only (Morgan and Truman 2022).

The severity of the offense was identified in prior research as a significant determinant of help-seeking generally (e.g., Brady et al. 2023; Fissel 2021; Gottfredson and Gottfredson 1990; and Reyns and Englebrecht 2010). Victims who experience more severe types of stalking are also more likely to engage in formal help-seeking (Ameral et al. 2020; Buhi et al. 2009; Fissel 2021; Jordan et al. 2007; and Reyns and Englebrecht 2014). According to Reyns and Englebrecht (2014), individuals who previously sought help, whether from formal or informal sources, are more likely to turn to the same type of source, formal or informal, when seeking help again. Lastly, studies indicated that stalking victims often hesitate to report the behavior to law enforcement if they know the perpetrator due to fear that their claims may not be taken seriously (Reyns and Englebrecht 2014). Bendlin and Sheridan (2019) hypothesized that the controlling behavior exhibited by stalkers can also leave victims feeling reluctant and fearful of disclosing the crime to the police.

### 2.3. Sexual Minority Help-Seeking

Individuals from marginalized groups, such as sexual minority individuals, may have lower levels of trust and confidence in the criminal justice system (Dario et al. 2020; Miles-Johnson 2013; and Owen et al. 2018). Studies examining violence against sexual minority individuals suggest that this group does not trust the police to treat them fairly and reported mistreatment by police officers (Dwyer and Ball 2012; and Finneran and Stephenson 2013). Additionally, sexual minority crime victims were shown to underutilize victim assistance services due to fear of secondary victimization by homophobic and insensitive workers and organizations (LaSala and Fedor 2020; and Messinger 2020). This lack of faith and trust in the system may influence the decision of sexual minority stalking victims to report their stalking victimization and/or seek help for their stalking victimization.

### 2.4. The Current Study

To date, there are few studies to investigate the factors that influence a stalking victim's help-seeking decisions in relation to their sexual orientation, despite research showing that sexual minority individuals experience disproportionally higher rates of stalking compared to heterosexual individuals (Edwards et al. 2015b, 2022; Langenderfer-Magruder et al. 2017; Sheridan et al. 2019; and Walters et al. 2013). The present study seeks to address this gap in the literature by investigating predictors of help-seeking, the influence of sexual orientation on help-seeking, and predictors of help-seeking across groups. Utilizing data from the United States-based 2019 NCVS-SVS, we will explore various demographic and psychological factors among both sexual minority and heterosexual victims of stalking; specifically, exploring if sexual minority and heterosexual stalking victims differ in factors that prevent them from seeking help.

### 3. Data and Methods

The current study employed data collected as part of the 2019 iteration of the Supplemental Victimization Survey of the NCVS-SVS. The United States Bureau of Justice Statistics (BJS) redesigned the 2006 Supplemental Victimization Survey (SVS) instrument in 2015 to align with the Violence Against Women Act (VAWA) updates from 2005 to 2013. The revamped SVS instrument included questions aimed at identifying each aspect of

VAWA's stalking definition, with expanded questions about unwanted contacts and behaviors related to traditional stalking and stalking with technology. Furthermore, additional questions were developed to assess the victim's fear and substantial emotional distress (Morgan and Truman 2022).

From June 2019 to December 2019, randomly sampled households with members 16 years or older, who passed initial screening questions for eligibility, were administered a one-time SVS after completing the primary National NCVS interview (Morgan and Truman 2022). Out of the original pool of 141,300 NCVS-eligible respondents, approximately 105,000 completed the SVS questionnaire, yielding a response rate of 74.3% (Morgan and Truman 2022). Eligibility for completing the SVS was contingent upon the NCVS-SVS definition, which is largely considered consistent with the United States federal definition of stalking (Morgan and Truman 2022). In order to be classified as a victim of stalking, it is necessary that the respondent experienced a repeated course of conduct that caused fear or substantial emotional distress or that would cause a reasonable person to experience such distress. The NCVS-SVS screener questions collected information on unwanted contacts or behaviors, repeated conduct, actual and reasonable fear, and substantial emotional distress.

The present study aims to explore individual factors that may contribute to sexual minority and heterosexual stalking victims' help-seeking behaviors. Therefore, the study was limited to NCVS-SVS respondents who responded to the sexual orientation questions asked during the NCVS. Starting in July 2019, the NCVS began asking respondents aged 16 and older questions about their sexual orientation only if they reported victimization (Flores et al. 2022; Morgan and Truman 2022). The NCVS captured sexual orientation by asking, "Which of the following best represents how you think of yourself?" The six response categories included: (1) gay (lesbian, for female respondents); (2) straight, that is, not gay (or lesbian); (3) bisexual; (4) something else; (5) I do not know the answer; and (6) refused. The responses were then dichotomized into heterosexual and sexual minority. This analysis does not include a variable for gender (e.g., cisgender, transgender, nonbinary, etc.). For this reason, "sexual minority" was preferred over LGBTQ+. After limiting the sample to just the respondents who responded to questions included in variables used in the present study, the analytic sample consisted of 1906 mostly heterosexual (n = 1767) individuals.[1]

### 3.1. Dependent Variable

A single dependent dichotomous measure of "sought help" was created for the current study to capture if a victim sought external help or advice in response to their stalking victimization (1 = any support sought, 0 = no). The measure was derived from if respondents answered yes to any three questions that captured (a) whether the victim asked for advice or help from friends or family, (b) whether the victim sought help or advice from any office or agency (other than the police) that assists victims of crime, (c) or if the victim or someone on their behalf called or contacted the police to report any of the unwanted contacts or behaviors. Of the sample, approximately one-third (33.74%) of respondents, regardless of sexual orientation, sought external help or advice in response to their stalking victimization.

### 3.2. Independent Variables

Prior research showed that demographic characteristics may influence victims' decisions to report and seek help for stalking victimization (Barrett and St. Pierre 2011; Fissel 2021; and Reyns and Englebrecht 2010, 2014). As such, the present study controlled for age, race, and sex in its analyses.[2] The age of the victim in the present study was captured through the NCVS's continuous variable (V3014), excluding any participants under the age of 18. The sex of the victim was captured through the NCVS's allocated sex dichotomous categorical variable (V3018), "male" and "female", with male (0 = male) being the reference category. Race was based upon a dichotomous recode of the race variable (V3023A) from the NCVS, which captured the respondent's race. Respondents were given 12 options, including "white only", "black only", "Asian only", "American Indian/Alaskan Native

only", and seven mixed race combinations. Respondents who identified as "white only" were within one category, which was used as the reference group, and all other response categories were collapsed into a single category of "non-white".

Finally, emotional impact (i.e., fear and distress) was captured through a dichotomous measure, which captured if a victim responded yes or no to experiencing either fear, distress, or both fear and distress because of the unwanted contacts or behaviors experienced (1 = any yes, 0 = no). The measure was derived from if respondents answered yes to either or both of the following two questions from the 2019 NCVS-SVS: "Did any of these unwanted contacts or behaviors make you fear for your safety or the safety of someone close to you?" and "Did any of these unwanted contacts or behaviors cause you substantial emotional distress?".

*3.3. Analytic Strategy*

Addressing the research questions central to this study requires two separate processes to determine whether differences exist between heterosexual and sexual minority respondents in seeking help for stalking victimization. We further explore this in terms of age, sex, race, and emotional impact corresponding to stalking victimization in the bivariate context with *t*-tests and chi-square analyses. Finally, we employ logistic regression analyses, one for the sexual minority subsample and one for the heterosexual subsample, to assess the correlations between select individual factors and seeking help for stalking victimization. It is important to note that the sexual minority subsample size (n = 139) and the heterosexual subsample size (n = 1767) differ significantly. As a consequence, the statistical power and robustness of the coefficients between these models are different. Given this discrepancy, we caution against direct comparisons between the coefficients from these two models as this could lead to misleading interpretations. Instead, each logistic regression model should be interpreted independently within the context of its specific sample characteristics and sizes.

## 4. Results

The broader sample of victims consisted of 1906 respondents, predominantly heterosexual (92.71%), white (84.52%), and female (61.75%), with an average age of 49.86 years (SD = 15.73).[3] Notably, 32.74% of respondents reported seeking help for stalking victimization, and 28.91% reported an emotional impact due to stalking. Sexual minority respondents (N = 139) had a higher proportion of victims who sought help for stalking victimization (43.17%) compared to heterosexual respondents (31.92%). The sexual minority sample respondents' mean age was 43.24 (SD = 14.59), 61.15% female, 82.73% white, and 40.29% reported experiencing an emotional impact due to stalking. Heterosexual respondents (N = 1767) mean age was 50.38 (SD = 15.80), 61.80% female, and 84.67% white, and 28.01% reported experiencing an emotional impact due to stalking. Descriptive statistics for the full sample and both sub-samples can be seen in Table 1.

Initial results find a statistically significant difference indicating that sexual minority individuals were more likely to seek help compared to heterosexual individuals ($\chi^2(1)$ = 7.40, $p < 0.01$). Regardless of sexual orientation, females were more likely to seek help compared to males ($\chi^2(1)$ = 106.33, $p < 0.001$; 41.46% of females vs. 18.66% of males), along with those who experienced an emotional impact ($\chi^2(1)$ = 519.13, $p < 0.001$; 71.14% vs. 17.12%). Further, in the full sample, respondents who sought help were younger on average compared to those who did not (t(1904) = 8.77, $p < 0.001$; M = 44.97 for help-seekers vs. M = 52.23 for non-help-seekers). In the full sample, there was a difference in help-seeking by race.

Given the preliminary findings, two-sample t-tests assuming unequal variances and chi-square tests were conducted to compare demographic differences of help seekers by sexual orientation (see Table 2). For age, there was a significant difference in the means (t(76.05) = 4.74, $p < 0.001$), with heterosexual respondents who sought help (M = 45.88, SD = 16.44) being older than sexual minority respondents who sought help (M = 36.42, SD = 14.51). Presence of emotional impact was approaching significance ($\chi^2(1)$ = 3.14,

$p < 0.10$), with sexual minority respondents who sought help more likely to report experiencing emotional distress compared to heterosexual victims who sought help (73.33% vs. 61.70%, respectively). There were no significant differences between sexual minority and heterosexual help-seekers with regards to sex or race.

**Table 1.** Descriptive statistics for full and sub-samples (N = 1906).

|  | Full Sample (N = 1906) | | Heterosexual Sample (N = 1767) | | Sexual Minority Sample (N = 139) | |
|---|---|---|---|---|---|---|
|  | **N** | **%** | **N** | **%** | **N** | **%** |
| Sought help |  |  |  |  |  |  |
| Yes | 624 | 32.74 | 564 | 31.92 | 60 | 43.17 |
| No | 1282 | 67.26 | 1203 | 68.08 | 79 | 56.83 |
| Sex |  |  |  |  |  |  |
| Female | 1177 | 61.75 | 1092 | 61.80 | 85 | 61.15 |
| Male | 729 | 38.25 | 675 | 38.20 | 54 | 38.85 |
| Race |  |  |  |  |  |  |
| White | 1611 | 84.52 | 1496 | 84.67 | 115 | 82.73 |
| Non-white | 295 | 15.48 | 271 | 15.34 | 24 | 17.27 |
| Emotional impact |  |  |  |  |  |  |
| Yes | 551 | 28.91 | 495 | 28.01 | 56 | 40.29 |
| No | 1355 | 71.09 | 1272 | 71.99 | 83 | 59.71 |
|  | **M** | **SD** | **M** | **SD** | **M** | **SD** |
| Age | 49.86 | 17.29 | 50.38 | 17.21 | 43.24 | 17.06 |
|  | (18–90) | | (18–90) | | (18–88) | |

**Table 2.** Bivariate comparisons of help-seeking among victims (N = 624).

|  | Sexual Minority Help-Seekers (N = 60) | | Heterosexual Help-Seekers (N = 564) | | |
|---|---|---|---|---|---|
|  | **N** | **%** | **N** | **%** | $\chi^2$ |
| Sex |  |  |  |  |  |
| Male | 14 | 23.33 | 122 | 21.63 | 0.09 |
| Female | 46 | 76.67 | 442 | 78.37 |  |
| Race |  |  |  |  |  |
| White | 50 | 83.33 | 464 | 82.27 | 0.04 |
| Non-white | 10 | 16.67 | 100 | 17.73 |  |
| Emotional Impact |  |  |  |  |  |
| Yes | 44 | 73.33 | 348 | 61.70 | 3.14 [+] |
| No | 16 | 26.67 | 216 | 38.30 |  |
|  | **M** | **SD** | **M** | **SD** | **t** |
| Age | 36.42 | 14.51 | 45.88 | 16.44 | 4.74 *** |

*** $p < 0.001$; [+] $p < 0.10$.

*Logistic Regression Models*

Refining the sample to just those identifying as a sexual minority, a logistic regression model was developed to further examine the relationship between seeking help for stalking victimization and a limited number of key potential correlates; including age, sex, race, and emotional impact (see Table 3). The model was statistically significant ($\chi^2(4) = 74.02$, $p < 0.001$), and explained 39.02% of the variation in sexual minorities seeking help for stalking victimization. Seeking help among sexual minority stalking victims was significantly associated with younger respondents (b = −0.058, $p < 0.001$), female respondents (b = 1.056, $p = 0.036$), and those who experienced an emotional impact (b = 3.074, $p < 0.001$). Race of the sexual minority respondent was not a significant predictor of help-seeking.

**Table 3.** Logistic regression predicting seeking help for stalking victimization.

| | Sexual Minority Respondents (N = 139) | | | | Heterosexual Respondents (N = 1767) | | | |
|---|---|---|---|---|---|---|---|---|
| | b | SE | P > z | OR | b | SE | P > z | OR |
| Age | −0.058 | 0.016 | <0.001 | 0.942 | −0.014 | 0.004 | <0.001 | 0.986 |
| Sex (0 = Male) | 1.056 | 0.503 | 0.036 | 2.876 | 0.941 | 0.133 | <0.001 | 2.563 |
| Race (0 = White) | −0.485 | 0.620 | 0.434 | 0.616 | 0.069 | 0.167 | 0.677 | 1.072 |
| Emotional impact | 3.074 | 0.524 | <0.000 | 21.631 | 2.310 | 0.127 | <0.001 | 10.076 |
| Constant | 0.281 | 0.749 | 0.705 | 1.32 | −1.473 | 0.226 | <0.001 | 0.229 |
| Pseudo $R^2$ | 0.3902 | | | | 0.2366 | | | |
| Model $\chi^2$ (4) | 74.02 | | | | 523.60 | | | |
| | $p < 0.001$ | | | | $p < 0.001$ | | | |

The odds ratio for age indicates that for each one-unit increase in age, there was a corresponding decrease of 0.942 in the odds of sexual minority respondents seeking help for stalking victimization, holding all other variables constant, suggesting that older age is associated with a lower likelihood of seeking help for stalking victimization. Female sexual minority respondents had 2.841 times as likely odds of seeking help for stalking victimization, holding all other variables constant. The odds ratio for emotional impact indicates that sexual minority respondents who noted an emotional impact due to stalking victimization had greater odds of seeking help for stalking victimization (OR = 21.631, $p < 0.001$), holding all other variables constant.

For reference, a separate logistic regression model was run using a refined sample of just those respondents identifying as heterosexual (see Table 3). This model aimed to elucidate the relationship between heterosexual respondents seeking help for stalking victimization and the same key potential correlates as used in the sexual minority model: age, sex, race, and emotional impact. It is crucial to note, however, that the coefficients of the sexual minority model and the heterosexual model are not directly comparable due to the significant difference in sample sizes. The heterosexual model was statistically significant ($\chi^2(4) = 523.60$, $p < 0.001$), and explained 23.66% of the variation in seeking help for stalking victimization. Seeking help among heterosexual stalking victims was significantly associated with younger respondents (b = −0.014, $p < 0.001$), female respondents (b = 0.941, $p < 0.001$), and those who experienced an emotional impact (b = 2.310, $p < 0.001$). Race of the respondent was not a significant predictor of help-seeking.

The odds ratio for age indicates that for each one-unit increase in age, there was a corresponding decrease of 0.986 in the odds of heterosexual stalking victims seeking help for stalking victimization, holding all other variables constant, suggesting that older age is associated with a lower likelihood of seeking help for stalking victimization. Female heterosexual respondents had 2.563 times as likely odds of seeking help for stalking victimization, holding all other variables constant. The odds ratio for emotional impact indicates that heterosexual respondents who noted an emotional impact due to stalking victimization, had greater odds of seeking help for stalking victimization (OR = 10.076, $p < 0.001$), holding all other variables constant.

## 5. Discussion

The aim of the present study was to explore the relationship between sexual orientation and the help-seeking behaviors of stalking victims using the United States-based National Crime Victimization Survey Supplemental Stalking Survey for the year 2019. The findings suggest that, within the sexual minority population, younger, female respondents who experienced an emotional impact due to stalking victimization are more likely to seek help following their experiences.

Consistent with prior literature (Englebrecht and Reyns 2011; Kuehner et al. 2012; Spitzberg 2002; Sheridan and Lyndon 2012; Spitzberg et al. 2010; and Tjaden et al. 2000), findings indicate sex disparities in help-seeking behaviors. Female respondents in the present sample, regardless of sexual orientation, were more likely to seek help than male

respondents. This disparity held when looking at sexual minority victims only. These findings align with those of Langenderfer-Magruder et al. (2020), who reported that female sexual minority victims reported stalking to the police at higher rates than male sexual minority victims. This may be due to societal norms that place a greater emphasis on women's safety, or it may reflect differences in the types of stalking behaviors experienced by men and women (Cupach and Spitzberg 2004). Historically, societal expectations around masculinity and the perception of vulnerability may deter some male victims from seeking help or disclosing their experiences (Kilmartin 1994; Kinsey et al. 1998; and Johnson 2005). Further, males may be more likely to experience online stalking or harassment, which may be perceived as less severe or less worthy of reporting compared to more physical forms of stalking (Reyns et al. 2012). This is further supported by data collected by the United States-based Centers for Disease Control and Prevention (CDC). The National Intimate Partner and Sexual Violence Survey (NISVS) also collects nationally representative data on interpersonal violence, albeit from a smaller sample (27, 571 in 2016/2017). Their latest reports found that approximately 56% of females, compared to 61% of males, experienced stalking behaviors through email, text message, or social media (Smith et al. 2022).

An important aspect of our findings is the role of age in help-seeking behaviors among stalking victims. Our results indicate that younger individuals within the sexual minority community are more likely to seek help than their older counterparts. This pattern held true for both sexual minority and heterosexual individuals. This is consistent with research suggesting that younger victims tend to be more proactive in seeking help and are more likely to report stalking to authorities (Reyns and Englebrecht 2014). It is possible that younger individuals are more aware of available resources and support systems, or perhaps they are more likely to perceive stalking as a serious issue warranting intervention. Further, these data were collected in 2019 and asked respondents to reflect on recent incidents.

Race was not a significant predictor of help-seeking among the full sample or sexual minority stalking victims, specifically. Some studies found that race is a significant predictor of help-seeking for stalking victims, with white women being more likely to seek assistance (Campbell and Moore 2011, using a college sample; Reyns and Englebrecht 2014, using the 2006 NCVS-SVS); however, it is important to note that these investigations did not account for the potential influence of sexual orientation. It is possible that the United States #MeToo movement, beginning in 2018, fostered an environment of openness with regards to gender-based victimization. This movement, which was initiated by an African-American woman, potentially had some form of impact on the likelihood of non-white victims of gender-based crimes seeking help following their victimization. Palmer and colleagues suggest that post #MeToo, African-American women were more likely to disclose incidences of abuse and harassment, compared to before the movement. The present study's findings may indicate that there are other overriding barriers to help-seeking that impact all sexual minority victims of stalking, irrespective of their race. Barriers could include fear of not being taken seriously (Reyns and Englebrecht 2014), fear of secondary victimization (LaSala and Fedor 2020; and Messinger 2020), or a lack of trust in the justice system (Miles-Johnson 2013), which might be exceptionally high among non-white sexual minority individuals.

The emotional impact (i.e., fear and distress) of stalking victimization also plays a significant role in the help-seeking behaviors of victims. Our findings demonstrate that individuals, regardless of sexual orientation, are more likely to seek help when they experience fear or distress due to stalking. These findings align with previous research demonstrating the strong association between emotional impact and help-seeking behavior among stalking victims (Jordan et al. 2003; Mullen et al. 2001; Randa et al. 2022; and Reyns and Englebrecht 2010). For example, Randa et al. (2022) suggest that victims of stalking may be relatively unaffected by stalking behaviors unless they cause worry or concern. This assertion underscores the influence of perceived harm on victims' decisions to implement help-seeking measures. In essence, victims experiencing heightened fear and distress are more inclined to acknowledge the severity of their situation and actively seek assistance. While not measured in the NCVS-SVS, the NISVS measures physical and

psychological symptoms between individuals who were and were not victims of stalking. Both men and women with a history of stalking victimization were almost twice as likely to experience frequent headaches, difficulty sleeping, difficulty concentrating and making decisions, and difficulty doing errands alone, compared to those without a history of stalking victimization.

*5.1. Limitations and Future Research*

While this study provides valuable insights, it has several limitations that should be acknowledged. First, our sample size consisting of predominantly white and heterosexual respondents posed a significant limitation, particularly concerning the representation of sexual minorities. A dramatically shrunken sample size, coupled with the small number of sexual minority participants who were stalking victims in this study, limits the generalizability of our findings to this subpopulation. However, the NCVS-SVS is currently the most recent and largest dataset of sexual minority victims. Future datasets, such as the NISVS, should actively include a more diverse range of participants and consider sexual orientation as a regular demographic variable. The NISVS never included sexual orientation, with the exception of their special topic 2010 data collection.[4] Including sexual orientation would allow for a comparison of nationwide datasets regarding victimization among this population. Understanding the unique experiences of sexual minority stalking victims is crucial for developing responsive support services and policies.

It is important to note that our study does not distinguish between intimate partner (IP) and non-intimate (non-IP) stalking. This aspect could serve as a potential limitation, given the differing dynamics and complexities of IP and non-IP stalking situations. For instance, sexual and racial minorities may be more likely to experience both IP and non-IP stalking (Fedina et al. 2020), which may in turn influence their help-seeking behaviors. Thus, while our findings provide valuable insights, a more nuanced understanding of stalking victimization could be gained from examining the role of the victim–offender relationship.

Another potential barrier to broader applicability is that our methodology could not adequately capture variations in sex and gender. To broaden the generalizability of these findings, future research should include larger and more diverse sample populations, including individuals from diverse sexual orientation backgrounds, sex, and gender identities, including those who identify as transgender, along with a more diverse racial and ethnic sample. Therefore, future research should be deliberate in diversifying the sampling techniques to capture these variations. The integration of intersectionality, looking at race, gender, sex, age, and sexual minority status in victimology research, could contribute to a more nuanced understanding of the experiences and needs of different victim subpopulations. This approach could be essential for promoting inclusivity and the development of comprehensive victim support and interventions. Moreover, the cross-sectional design of our study restricts the depth of insight that can be gained from the results. Here, it is important to highlight that the NCVS, the foundation of our data source, uses a 12-month recall timeframe (Morgan and Truman 2022). Therefore, understanding that help-seeking is a process, which may not take place within this recall period, is essential. This knowledge may influence the interpretation and application of a number of theoretical frameworks that can prove useful in studying victim behavior and responses. Future studies employing longitudinal designs would provide a more in-depth understanding of stalking's long-term effects on victims and their help-seeking behaviors.

Notably, our present study utilized an aggregate measure of "sought help", which combined different forms of help-seeking, such as seeking help from friends or family, the police, or other offices or agencies. This consolidation, while effective for our analysis given the sample size, maybe did not capture distinct experiences or predictors associated with these different help-seeking routes. Particularly for diverse victim groups, such as heterosexual and sexual minority respondents, these forms of help-seeking could vary significantly. Future research, given larger sample sizes, could benefit from a more granular approach, breaking down the "sought help" measure into its constituent parts. This would



afford a more nuanced understanding of victims' help-seeking behaviors and the factors influencing these decisions.

The present study was unable to provide detailed information on why victims did or did not seek help, which could obscure critical insights into barriers to help-seeking among stalking victims. To further explore factors that may impact help-seeking behaviors among stalking victims, future research may consider employing multiple measures of help-seeking and incorporate qualitative methods to develop a more comprehensive understanding of the barriers and motivations associated with seeking or not seeking help. For instance, including questions related to property damage, threats, physical attacks, offender characteristics (such as the number of offenders, their sex, age, race/ethnicity, and relationship to the victim), and situational characteristics (such as duration, frequency, type of discovery, and stalker's motive) can provide valuable insights into the complex dynamics of help-seeking among stalking victims. Furthermore, future research should investigate the factors influencing help-seeking among sexual minority victims of traditional stalking compared to cyberstalking to develop tailored interventions and support services. Such data could prove instrumental in tailoring interventions to address these barriers and better support stalking victims.

Finally, whereas the current study primarily focused on personal characteristics and how they informed victim decision-making, future researchers may find it fruitful to explore theoretical explanations for sexual minority help-seeking post-victimization. There are a number of theoretical frameworks that may prove useful in this regard. For example, Liang et al.'s (2005) cognitive theory, which they utilized in the context of domestic violence help-seeking, emphasizes individual, interpersonal, and sociocultural influences on decision-making. Similarly, the focal concerns theory and criminal justice actor decision-making frameworks of Steffensmeier et al. (1998) and Gottfredson and Gottfredson (1990), respectively, were successfully utilized to better understand the decision-making in stalking cases (e.g., Brady 2023; Brady and Reyns 2020; and Brady et al. 2020).

*5.2. Policy Recommendations*

Findings from the present study suggest that individual characteristics, such as sex, age, and the emotional impact of stalking, significantly influence help-seeking behavior among sexual minority stalking victims. While these individuals were more likely to seek help, they also were more likely to experience an emotional impact from their victimization, compared to heterosexual victims of stalking. These findings highlight the need for targeted support and resources that consider these factors. Based on these findings, we propose several policy recommendations.

Public awareness campaigns are suggested to educate about the effects of stalking, emphasize the importance of seeking help, and challenge the stigma associated with stalking victimization. These campaigns should specifically target misconceptions or stigma prevalent in vulnerable populations such as sexual minority individuals (Weller et al. 2013). Additionally, general resources, such as hotlines, counseling services, support groups, and tailored services for sexual minority stalking victims, could offer significant assistance. These specialized resources could provide more relatable and effective support for this community. Furthermore, collaborations between government agencies, non-profit organizations, and community groups could create comprehensive support systems for stalking victims. Meaningful partnerships with community-based organizations, especially those advocating for the rights and well-being of sexual minority individuals, could facilitate the development of more effective, inclusive, and responsive interventions and policies by pooling resources and knowledge (Craig et al. 2015).

Enhancing police training on sexual minority issues, including sexual minority components in trauma-informed training (Levenson et al. 2023), establishing sexual minority liaison officers (Pickles 2020), promoting inclusive victim assistance services (Meyer 2015), implementing anti-discrimination policies (Nakamura et al. 2022), and collaborating with sexual minority organizations (Craig et al. 2015), can bridge the trust gap between marginal-

ized communities and the criminal justice system, encouraging more victims to seek help. Given the sex disparities in help-seeking behaviors identified in our study, targeted actions to address the barriers males face in reporting stalking incidents are crucial. It is imperative to ensure that trauma-informed training and victim assistance services explicitly address the unique challenges faced by sexual minority individuals, creating a supportive and inclusive environment for all victims.

Additionally, age-specific interventions should be considered, given the varying help-seeking behaviors across different age groups. Different age groups may have varying levels of awareness about available resources and have unique needs due to their life experiences. Hence, efforts should be made to make victim services more widely known and accessible while ensuring accessibility to both physical and virtual support spaces, particularly for marginalized groups such as sexual minority individuals. As Morgan and Truman's (2022) findings suggest, a majority of stalking victims who sought victim services in the United States in 2019 received help, demonstrating that these services can provide effective support to victims. Future policy developments should consider the specific factors that influence help-seeking behavior among stalking victims. By doing so, we can better facilitate these behaviors, improve interventions, and enhance support services, thus promoting safety and well-being.

Of course, these suggestions are reactive responses to stalking victimization. Proactively, educational programming can help decrease perpetration of various forms of interpersonal violence, ideally including stalking behaviors. Prevention programs traditionally tend to be directed toward males (Graham et al. 2019), based on the higher likelihood of male perpetrators for both male and female victims of stalking (Smith et al. 2022). Further, prevention programs are directed at interpersonal violence generally, and not just stalking. Stalking is an especially difficult concept to educate adolescents on, due to the closeness of stalking behaviors with common adolescent dating behaviors (Theriot 2008). For this reason, it is imperative that stalking is added to dating violence curriculums, beginning with adolescents and continuing as part of required incoming student training for higher education students. A relatively new, but promising program for higher education settings is Catharsis, You Got This! This evidence-based online training covers a range of interpersonal violence, including bystander intervention and stalking, and also incorporates diverse populations and views, highlighting different racial and ethnic backgrounds, religious beliefs, and sexual orientations (Fansher and Zedaker 2022). The latter is important as gender-based prevention programs often ignore experiences unique to sexual minority individuals (Kettrey and Marx 2019). While programs such as You Got This! will need to be empirically evaluated, programs that challenge gender abuse myths, encourage bystander intervention (Hooker et al. 2020), and include diverse populations and lifestyles will arguably only improve the reach and impact of prevention efforts.

**Author Contributions:** Conceptualization: J.C.F., A.K.F., R.R. and B.W.R.; Methodology: J.C.F., A.K.F. and R.R.; Formal analysis: J.C.F.; Resources: J.C.F. and A.K.F.; Writing—original draft: J.C.F. and B.W.R.; Writing—review & editing: A.K.F., R.R. and B.W.R.; Visualization: J.C.F.; Supervision: R.R. All authors have read and agreed to the published version of the manuscript.

**Funding:** This research received no external funding.

**Institutional Review Board Statement:** The study was conducted in accordance with the Declaration of Helsinki, and approved by the Institutional Review Board of Sam Houston State University, Huntsville, TX, USA (Approval number: IRB-2023-129, approved on 5 May 2023).

**Informed Consent Statement:** Data from this research was analyzed solely as secondary data.

**Data Availability Statement:** Data available in a publicly accessible repository. The data presented in this study are openly available online through the Inter-university Consortium for Political and Social Science Research (ICPSR) at https://doi.org/10.3886/ICPSR37950.v1 (accessed on 5 May 2023).

**Conflicts of Interest:** The authors declare no conflict of interest.

## Notes

[1] Approximately 15,000 respondents did not answer the question about sexual orientation. The remaining loss concerns the help-seeking questions, which were missing from approximately 76,000 cases.

[2] Our selection of independent variables adhered to the 'Events per Variable' (EPV) principle, a respected standard in statistical analysis that recommends having at least 10 instances of the least common outcome for each independent variable (Peduzzi et al. 1996). In our model, the least occurring outcome was sexual minority stalking victims seeking help, with a count of 60 out of 139 observations (0.43 probability). For instance, our model has 5 independent variables, and the estimated probability of the least frequent outcome is 0.43; the minimum required sample size is approximately 116 (obtained by multiplying 10 by 5 and dividing by 0.43, rounding up to the nearest whole number). Indicating that our sample size exceeds the minimum requirement, ensuring a robust foundation for estimating the logistic regression coefficients and obtaining statistically significant results.

[3] In this sample, 12 heterosexual individuals' ages were top coded to 90 per the Bureau of Justice Statistics (BJS) confidentiality procedures. Analyses run, with and without these participants, showed no significant statistical difference; therefore, they were retained in the sample.

[4] A newer round of of NSIVS data collection captured sexual orientation, but the report was not available for comparison as of this writing.

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
