# Peer review of "Reporting and Help-Seeking among Sexual Minority and Heterosexual Victims of Stalking"

_socsci, doi:10.3390/socsci12080424_

Round 1

Reviewer 1 Report

Some mistakes: Line 72 you should say among instead of mong; line 277 heter-, should be hetero-. 

It is really intersting the way you explain stalking but no enough to explain the heterosexual minority as opression group to foster patriachal society, heterocestrims as the way of violence towards other -maybe better- sexual desires. They are the offenders of freedom and sexual rigths.  According to research Heterosexality and Homosexualidty are minorities group in the sacale of a clasical USA research Alfred Kinsey (1947 & 1953). 

You need to foster also from my view than improve the legal regulation is not emough if we do not develope Conflict Resolution Programmes and Gender Violence Prevention in all educational levels as well as.

Besides this recomendations I consider you have done an excellent work and all the introduction, methods, results and conclusions are really very good.

It is an excellent work

Author Response

R1 C1:

Some mistakes: Line 72 you should say among instead of mong; line 277 heter-, should be hetero-. 

We appreciate the reviewers attention to detail and have corrected the listed typos.

R1

C2:

It is really intersting the way you explain stalking but no enough to explain the heterosexual minority as opression group to foster patriachal society, heterocestrims as the way of violence towards other -maybe better- sexual desires. They are the offenders of freedom and sexual rigths.  According to research Heterosexality and Homosexualidty are minorities group in the sacale of a clasical USA research Alfred Kinsey (1947 & 1953). 

We appreciate the author’s attention to this issue. We have included reference to Kinsey’s work within the text and hope that this is satisfactorily addresses the reviewer’s comment.

R1

C3:

You need to foster also from my view than improve the legal regulation is not emough if we do not develope Conflict Resolution Programmes and Gender Violence Prevention in all educational levels as well as.

We agree, primary prevention is extremely important. We have added a large paragraph to the policy implications addressing this.

Reviewer 2 Report

Overall, this is a strongly written manuscript using the most recent data addressing stalking within the National Crime Victimization Survey. The authors attempt to understand help-seeking differences among and between heterosexual and sexual minority individuals that experienced stalking. There are a few areas where the manuscript can be strengthened or clarified that a are detailed below.

Methods/Limitations: need to make clear that NCVS uses a 6-month recall timeframe and that help-seeking is a process in that it may not take place within the recall period of the survey (this also underlines the point regarding the need for longitudinal research; also see comment about help-seeking theory later)

Methods/Analysis:

Justification for sample size is needed and some note of power/effect size within the model given these issues—increase confidence in the data with a bit more. It may also be helpful to note that weighting is not recommended with these data (NCVS, when doing predictive and exploratory analyses).

Lit Review/Discussion

One thing missing is any discussion of help-seeking theory or frameworks. There are several that are applicable and positioning some of the literature review and discussion in this context will strengthen the paper. One that is often used for domestic violence and maps onto stalking well given the pattern of victimization is Liang et al. (2005).

Discussion

-Bring in CDC report more into the discussion (Smith, 2022) It has relevance to several points in the discussion (e.g. experiences for men v. women). It is also important to point out that the most recent population based stalking studies were NCVS supplement and the NISVS, therefore maximizing comparisons where possible.

-I was unclear of the of the relevance of noting the MeToo movement—it does not seem to fit well within the narrative and was not cited. It relates to sexual violence specifically so was trying to understand the connection to this paper.  If included, there have been a few articles regarding its impact on help-seeking that should be cited. One alternative is to instead focus more of the discussion on enhancing the comparison between NCVS and NISVS data and discussing the need for BJS to integrate stalking into their redesign or offer SVS more regularly to gain better data.

-It needs to be stated that you did not account for victim-offender relationship and that intimate partner stalking does differ from non-intimate stalking. This is a large part of the literature that should be noted and put into context within the study’s findings. For example, sexual and racial minorities may be more likely to experience both IP and non-IP stalking. Consider including to add to this point in the discussion - Fedina et al. (2020) Prevalence and sociodemographic factors associated with stalking victimization among college students, Journal of American College Health, 68:6, 624-630, DOI: 10.1080/07448481.2019.1583664

Author Response

R2 C1:

Methods/Limitations: need to make clear that NCVS uses a 6-month recall timeframe and that help-seeking is a process in that it may not take place within the recall period of the survey (this also underlines the point regarding the need for longitudinal research; also see comment about help-seeking theory later)

We thank the reviewer for the valuable suggestion. We have incorporated the clarification regarding the National Crime Victimization Survey's (NCVS) use of a 12-month recall timeframe in the most recent data collection to the limitations and future research section. This addition ensures transparency and acknowledges the potential limitations associated with the specific recall period of the survey. Furthermore, we appreciate your emphasis on help-seeking as a process that may extend beyond the recall period of the survey. This highlights the need for longitudinal research to capture the dynamic nature of help-seeking behaviors over time. We have incorporated this point in the discussion of limitations, emphasizing the potential benefits of longitudinal studies in understanding the complexities of help-seeking among stalking victims.

R2 C2

Methods/Analysis:

Justification for sample size is needed and some note of power/effect size within the model given these issues—increase confidence in the data with a bit more. It may also be helpful to note that weighting is not recommended with these data (NCVS, when doing predictive and exploratory analyses).

We appreciate the reviewer's suggestion regarding the need for justification of the sample size and the inclusion of information on power/effect size within the model. We have addressed this concern by incorporating a footnote (Footnote 2) in the Methods section that explains the adherence to the "Events per Variable" (EPV) principle and its role in ensuring an appropriate sample size. This addition provides the necessary justification for the sample size and increases confidence in the data.

R2 C3

Lit Review/Discussion

One thing missing is any discussion of help-seeking theory or frameworks. There are several that are applicable and positioning some of the literature review and discussion in this context will strengthen the paper. One that is often used for domestic violence and maps onto stalking well given the pattern of victimization is Liang et al. (2005).

We have added a paragraph to the discussion section that addresses potential theoretical frameworks, including Liang et al., as suggested by the reviewer.

Discussion

R2 C4

Bring in CDC report more into the discussion (Smith, 2022) It has relevance to several points in the discussion (e.g. experiences for men v. women).

·       - It is also important to point out that the most recent population based stalking studies were NCVS supplement and the NISVS, therefore maximizing comparisons where possible.

Thank you for this suggestion. Unfortunately, the most recent NISVS did not measure sexual orientation or if the victim sought help. We did add comparisons for experiencing online stalking and also potential physical and psychological consequences of stalking. Further, we included the need for more inclusive variables in the limitations and future research.

R2 C5

I was unclear of the of the relevance of noting the MeToo movement—it does not seem to fit well within the narrative and was not cited. It relates to sexual violence specifically so was trying to understand the connection to this paper.  If included, there have been a few articles regarding its impact on help-seeking that should be cited.

-One alternative is to instead focus more of the discussion on enhancing the comparison between NCVS and NISVS data and discussing the need for BJS to integrate stalking into their redesign or offer SVS more regularly to gain better data.

Thank you for this suggestion. We have removed this language from the discussion of age and have added a citation supporting this based on race.

The policy and future research, note the need of the NISVS to include more information on sexual orientation for a better comparison between the two datasets.

R2 C6

-It needs to be stated that you did not account for victim-offender relationship and that intimate partner stalking does differ from non-intimate stalking. This is a large part of the literature that should be noted and put into context within the study’s findings. For example, sexual and racial minorities may be more likely to experience both IP and non-IP stalking. Consider including to add to this point in the discussion - Fedina et al. (2020) Prevalence and sociodemographic factors associated with stalking victimization among college students, Journal of American College Health, 68:6, 624-630, DOI: 10.1080/07448481.2019.1583664

We appreciate the reviewer's insight regarding the importance of accounting for the victim-offender relationship and acknowledging the differences between intimate partner stalking and non-intimate stalking. We have incorporated this consideration into the limitations and future research section of the manuscript. We also thank the reviewer for the Fedina et al., (2020) citation suggestion, which we have incorporated as well.

Reviewer 3 Report

This study uses data from the NCVS Supplemental Victimization Survey to examine help seeking among victims of stalking, with a particular focus on victims who identify as a sexual minority. The topic is interesting and important and the manuscript is generally well written. However, I have some concerns about the analyses that should be addressed before the study is suitable for publication.

The dependent variable is an overall measure of “sought help” that is made up of multiple types of help seeking (friends/family, police, or other office/agency). I recommend examining these 3 items separately, as they likely have different predictors and, more importantly, the use of these could vary across heterosexual versus sexual minority respondents.

Table 3 estimates a logistic regression model predicting help seeking among the subsample of sexual minority respondents. I recommend adding a model predicting help seeking among the subsample of heterosexual respondents as well, in order to determine whether the predictors among sexual minority respondents are different. It would also be useful to conduct this analysis with the full sample, including the sexual minority versus heterosexual variable. This would allow us to see whether sexual minority respondents were more or less likely to seek help than heterosexual respondents after controlling for other relevant predictors.

Some important predictors of help seeking were not included as predictors. In addition to the fear/distress variable, the NCVS SVS appears to have included questions about whether the stalking led to property damage, threats, or physical attacks. There were also questions about offender characteristics (number of offenders, sex, age, race/ethnicity, relationship to the victim, etc.) and situational characteristics (duration, frequency, type of discovery, stalker’s motive, etc.). I recommend incorporating these into the analyses.

Be sure to consistently use terms. For example, use either “emotional impact” or “fear and distress” in the tables and in the text.

In the introduction, go ahead and state the year that the most recent NCVS SVS was collected.

Author Response

R3 C1:

The dependent variable is an overall measure of “sought help” that is made up of multiple types of help-seeking (friends/family, police, or other office/agency).

·       I recommend examining these 3 items separately, as they likely have different predictors and, more importantly, the use of these could vary across heterosexual versus sexual minority respondents.

We appreciate the reviewer's suggestion to examine the individual types of help-seeking (friends/family, police, or other office/agency) separately, as they may have distinct predictors and could vary across heterosexual and sexual minority respondents. However, due to the limitations of our sample size, we were unable to analyze these three items separately in the current study.

Nevertheless, we recognize the importance of this suggestion and its potential to provide valuable insights into the nuanced help-seeking patterns among different groups. We have incorporated this recommendation into the “Limitations and Future Research” section of the manuscript, highlighting the need for future studies to explore the individual types of help-seeking and their predictors, specifically considering the differences between heterosexual and sexual minority respondents.

R3

C2:

Table 3 estimates a logistic regression model predicting help seeking among the subsample of sexual minority respondents.

1.     I recommend adding a model predicting help seeking among the subsample of heterosexual respondents as well, in order to determine whether the predictors among sexual minority respondents are different.

2.      It would also be useful to conduct this analysis with the full sample, including the sexual minority versus heterosexual variable. This would allow us to see whether sexual minority respondents were more or less likely to seek help than heterosexual respondents after controlling for other relevant predictors.

Thank you for your insightful suggestions regarding the logistic regression analysis. We appreciate your interest in examining the differences between sexual minority and heterosexual respondents in terms of help-seeking behaviors.

We have taken your suggestion into consideration and have added a model predicting help-seeking among the subsample of heterosexual respondents to Table 3 that can be used as a reference for those interested. However, due to space constraints and the scope of the current study we did not add in the full sample logistic regression results.

R3

C3:

Some important predictors of help seeking were not included as predictors. In addition to the fear/distress variable, the NCVS SVS appears to have included questions about whether the stalking led to property damage, threats, or physical attacks. There were also questions about offender characteristics (number of offenders, sex, age, race/ethnicity, relationship to the victim, etc.) and situational characteristics (duration, frequency, type of discovery, stalker’s motive, etc.). I recommend incorporating these into the analyses.

We appreciate the reviewer's input on including additional predictors in our analyses, such as property damage, threats, physical attacks, offender characteristics, and situational characteristics. These variables offer valuable insights into the dynamics of stalking victimization and help-seeking.

While we were unable to include these variables in our current model due to sample size constraints we acknowledge the value of their suggestions in enhancing our understanding of help-seeking behaviors among stalking victims. We have incorporated these recommendations into the limitations and future research section of our manuscript, recognizing the exclusion of these predictors as a limitation of our study. By addressing these variables, future studies can provide a more nuanced understanding of help-seeking behaviors among stalking victims and inform the development of targeted interventions to support and empower individuals experiencing stalking victimization.

R3

C4:

Be sure to consistently use terms. For example, use either “emotional impact” or “fear and distress” in the tables and in the text.

We thank the reviewer for their attention to detail. We have addressed the issue of terminology consistency and made the necessary updates throughout the manuscript. Both the tables and the text now consistently utilize the term "emotional impact" to ensure uniformity and clarity.

R3

C5:

In the introduction, go ahead and state the year that the most recent NCVS SVS was collected.

Thank you for your suggestion. We have revised the introduction to include the specific year of data collection for the most recent NCVS SVS, which is 2019. This addition provides clarity regarding the timeframe of the survey data utilized in our study.

Round 2

Reviewer 3 Report

The revisions have not adequately addressed my concerns with the study.

Author Response

After revisiting reviewer 3's comments, we maintain our belief that we have addressed their feedback as comprehensively as our study's constraints permit.    While their suggestions to add more variables and analyze individual types of help-seeking were insightful, our data constraints meant we could only reflect upon these in the "Limitations and Future Research" section. We also incorporated a model for help-seeking among heterosexual respondents in Table 3, while other suggested predictors were acknowledged but not included due to the same limitations. Additionally, we have ensured consistent terminology and have included the year of data collection as per the reviewer's feedback in the revised manuscript.   Given our revisions/responses to reviewer 3's initial comments, we believe our current manuscript best represents our research within its limitations. Therefore, we respectfully submit our current manuscript without further revisions.